# Dietary Fibre Intake in Relation to Asthma, Rhinitis and Lung Function Impairment—A Systematic Review of Observational Studies

**DOI:** 10.3390/nu13103594

**Published:** 2021-10-14

**Authors:** Emmanouela Sdona, Athina Vasiliki Georgakou, Sandra Ekström, Anna Bergström

**Affiliations:** 1Institute of Environmental Medicine, Karolinska Institute, 171 77 Stockholm, Sweden; athina.georgakou@ki.se (A.V.G.); sandra.ekstrom@ki.se (S.E.); anna.bergstrom@ki.se (A.B.); 2Centre for Occupational and Environmental Medicine, Region Stockholm, 113 65 Stockholm, Sweden

**Keywords:** diet, fibres, asthma, rhinitis, respiratory test, obstructive lung disease

## Abstract

A high intake of dietary fibre has been associated with a reduced risk of several chronic diseases. This study aimed to review the current evidence on dietary fibre in relation to asthma, rhinitis and lung function impairment. Electronic databases were searched in June 2021 for studies on the association between dietary fibre and asthma, rhinitis, chronic obstructive pulmonary disease (COPD) and lung function. Observational studies with cross-sectional, case–control or prospective designs were included. Studies on animals, case studies and intervention studies were excluded. The quality of the evidence from individual studies was evaluated using the RoB-NObs tool. The World Cancer Research Fund criteria were used to grade the strength of the evidence. Twenty studies were included in this systematic review, of which ten were cohort studies, eight cross-sectional and two case–control studies. Fibre intake during pregnancy or childhood was examined in three studies, while seventeen studies examined the intake during adulthood. There was probable evidence for an inverse association between dietary fibre and COPD and suggestive evidence for a positive association with lung function. However, the evidence regarding asthma and rhinitis was limited and inconsistent. Further research is needed on dietary fibre intake and asthma, rhinitis and lung function among adults and children.

## 1. Introduction

Epidemiologic evidence has consistently shown that a high intake of dietary fibre is associated with a reduced risk of several chronic diseases, such as cardiovascular diseases, cancer, type 2 diabetes and obesity, as well as of total and specific-cause mortality [1,2,3]. Fibre-rich, plant-based dietary patterns, including grains, fruits, vegetables and nuts, stimulate the growth of beneficial bacterial species and contribute to a healthy colonic microbiota ecosystem due to the fermentation of fibres into short-chain fatty acids (SCFAs) [4].

Asthma is a chronic inflammatory disorder of the airways and the most common chronic disease among children. It is a cause of substantial burden of a disease, including a reduced quality of life in people of all ages and premature death [5]. Children with asthma, particularly those with persistent and severe forms of asthma, may attain a lower maximum lung function in adulthood, which increases the risk for the development of chronic obstructive pulmonary disease (COPD) [6]. Additionally, asthma frequently coexists with rhinitis, mostly among adolescents, as well as other atopic diseases, and it has been suggested that allergy-related diseases cannot be studied as isolated entities [7]. Both genetic and environmental factors have been implicated in the aetiology of the aforementioned diseases; however, the increase in the prevalence of asthma and other allergic diseases in the second half of the 20th century has been mostly associated with environmental factors, such as smoking, air pollution and changes in lifestyle and diet [8]. Following this increase, an increasing interest in identifying potentially modifiable factors has been expressed in the literature.

In recent years, epidemiological studies have also explored the association between dietary fibres and respiratory and allergic diseases. Dietary fibres may influence the development of respiratory and atopic outcomes through different mechanisms—for example, through the antioxidant and anti-inflammatory effects of whole grains, by enhancing the bio-accessibility of antioxidants from fruits and vegetables or through immunomodulatory effects induced by changes in the gut microbiota [9,10,11]. However, the epidemiological evidence for this potential association has not been systematically reviewed.

The aim of this systematic review is, therefore, to explore the existing evidence on dietary fibre intake in relation to asthma, rhinitis, COPD and lung function.

## 2. Materials and Methods

### 2.1. Protocol and Registration

This systematic review was performed according to the PRISMA guidelines [12] (checklist in the Appendix A), and an application for registration in PROSPERO was submitted.

### 2.2. Eligibility Criteria

Original studies reporting empirical findings on the association between dietary fibre intake and at least one outcome of interest (asthma; rhinitis; COPD, or symptoms of the aforementioned diseases, such as wheeze, cough, and phlegm; lung function) were searched. Observational studies on humans with cross-sectional, case–control or prospective designs were included. Studies on animals, case studies (case reports or case series) and intervention studies were excluded.

### 2.3. Information Sources

Systematic searches using predefined search terms were performed in multiple databases, including Medline (OVID), Embase, Cochrane Library, Web of Science and Scopus. The databases were searched from inception, limited to the English, French, German and Swedish languages. Additionally, reference lists of the articles included in the review and of relevant review studies were manually screened to identify other relevant articles. Information from conference abstracts, dissertations and grey literature (e.g., reports) was not included.

### 2.4. Search Strategy

The search was conducted in June 2021 based on the term construct used for Medline (see the Appendix A), assisted by professional librarians at the Karolinska Institute University Library. The following MeSH terms were used in the Medline (OVID) search: Dietary Fiber, Lung Diseases, Obstructive, Rhinitis and Respiratory Function Tests. The MeSH terms were adapted in accordance with the corresponding vocabulary in Embase Emtree. Each concept was also complemented with relevant free-text terms. The free-text terms were, if appropriate, truncated and/or combined with proximity operators. The full search strategies are available in the Appendix A.

### 2.5. Study Selection

The search results were exported to Endnote X9, where duplicates were excluded. As the first step, relevant articles were considered based on their title and abstract. At the second step, full-text versions of the selected papers were examined. In case there were multiple publications from the same cohort study, they were all included if they referred to different outcomes of interest. Following the above inclusion and exclusion criteria, two reviewers (E.S. and A.V.G.), without consideration for the results, performed the assessment of the studies for potential inclusion independently. Any differences in opinions were resolved through discussion until a consensus was reached. A third reviewer (S.E.) was consulted when necessary.

### 2.6. Data Extraction

The two reviewers independently conducted the data extraction from each study using a predefined data extraction sheet. The items extracted regarding the study characteristics comprised the first author name; year of publication; objectives; country; name of cohort (if applicable); study design; sample size; source population (age, sex and other characteristics); exposure assessment; categorisation of exposure; outcome assessment; mean follow-up period (if applicable); statistical methods; effect measures; covariates; missing data; control for selection bias and confounding, effect modifications and sensitivity analyses.

### 2.7. Risk of Bias in Individual Studies

The ‘Risk of Bias for Nutrition Observational Studies’ (RoB-NObS) tool recently developed by the US Department of Agriculture (USDA) Nutrition Evidence Systematic Review (NESR) team [13] was used to assess the risk of bias in individual studies independently by two reviewers (E.S. and A.V.G.) in the following domains: bias due to confounding, selection bias, bias in the classification of exposures, bias due to departures from intended exposures, bias due to missing data, bias in the measurement of outcomes and bias in the selection of reported results. On the occasion of discrepancies, a third reviewer (S.E.) assessed the study, and a consensus was achieved.

### 2.8. Presentation/Synthesis of Results

We performed a qualitative synthesis of the results, including a summary table presenting the association between the total fibre intake and the outcomes of interest (highest vs. lowest category and *p* for trends) from each study. Associations between different sources of fibre and the outcomes, as well as the stratified analysis results, are reported in text only.

### 2.9. Risk of Bias across Studies

The World Cancer Research Fund (WCRF) criteria [14], applied as suggested by Arnesen et al. [15], were used to grade the strength of the evidence for each outcome of interest as convincing (high), probable (moderate), limited/suggestive (low) and limited/no conclusion (insufficient).

## 3. Results

### 3.1. Study Selection

A flowchart of the study selection is presented in Figure 1. Briefly, the search of the electronic databases yielded 1328 articles, 169 of which were considered relevant after title and abstract screening. Additionally, one article was identified in the reference lists. Finally, 20 articles were considered for inclusion in the systematic review. Of these, ten were cohort studies, eight cross-sectional studies and two case–control studies. Ten studies were published the last five years (2017–2021) and ten studies before 2017 (2004–2016). Seven studies were conducted in the US, seven studies in Asia, three studies in Europe and three studies in Australia.

### 3.2. Study Characteristics

One birth cohort study explored fibre intake during pregnancy [16], two studies explored fibre intake during childhood [17,18] and 17 studies explored fibre intake during adulthood [19,20,21,22,23,24,25,26,27,28,29,30,31,32,33,34,35] (Table 1). Sixteen studies were population-based [18,19,20,21,22,23,24,25,26,27,28,29,31,32,33,34], while four studies were conducted on high-risk populations (infants with a family history of allergic disease [16], migrant adolescents [17], COPD adults [30] or smokers [35]). Half of the studies included fibre intake as the primary exposure [16,19,20,22,24,26,27,28,29,32], while the other half studied fibre intake as part of the nutrient or food intake or dietary pattern [17,18,21,23,25,30,31,33,34,35]. With regards to the outcomes of interest, seven studies reported associations with asthma/asthma symptoms [16,17,18,19,20,22,23], two studies with rhinitis [20,21], eight studies with COPD/COPD symptoms [24,25,26,27,28,29,30,31] and six studies with lung function [23,24,32,33,34,35] (seven studies reported more than one outcome).

### 3.3. Results of Individual Studies

#### 3.3.1. Maternal Fibre Intake during Pregnancy

An Australian cohort study of 639 mother–infant pairs, including infants with a family history of allergic disease, found that, although the total fibre intake during pregnancy was not associated with allergic disease in the offspring, a higher resistant starch intake was associated with a reduced risk of infant wheeze up to age 12 months (OR 0.68; 95% CI 0.49–0.95) [16].

#### 3.3.2. Fibre Intake during Childhood

Two cross-sectional studies reporting fibre intake during childhood were identified, with inconsistent results. An Australian study of 144 adolescents aged 12–18 years reported no association between fibre intake and self-reported wheeze [17]. However, a study among 4133 children aged 2–11 years from the US National Health and Nutrition Survey (NHANES) indicated that the odds of having asthma were higher for children who had a lower fibre intake (Q1 vs. Q4, ever asthma OR 1.31; 95% CI 0.88–1.96, *p*-trend 0.034 and current asthma OR 1.38; 95% CI 0.87–2.20, *p*-trend 0.027) [18]. In this study, the median fibre intake was 6.7 g/1000 kcal, which is below the recommended US intake of 14 g/1000 kcal.

#### 3.3.3. Fibre Intake during Adulthood

##### Asthma, Rhinitis and Related Symptoms

Five studies reporting fibre intake during adulthood in relation to asthma, rhinitis and related symptoms were identified, with somewhat consistent results. In a cross-sectional study of 13,147 adults from the US NHANES, a low fibre intake was associated with increased odds of prevalent asthma (Q1 vs. Q4, OR 1.4; 95% CI 1.0–1.8, *p*-trend 0.092), wheeze (OR 1.3; 95% CI 1.0–1.6, *p*-trend 0.017), cough (OR 1.7; 95% CI 1.2–2.3, *p*-trend < 0.001) and phlegm (OR 1.4; 95% CI 1.1–2.0, *p*-trend 0.011) [19]. Regarding asthma, stronger associations were seen for women and for non-Hispanic White adults. In a cross-sectional study of 10,479 adults from the Korean National Health and Nutrition Examination Survey (KNHANES), a higher dietary fibre intake was associated with reduced odds of asthma (Q4 vs. Q1, OR 0.66; 95% CI 0.48–0.91, *p*-trend < 0.001) and allergic rhinitis, the latter, however, only for Q2 vs. Q1 (OR 0.84; 95% CI 0.70–1.00, *p*-trend < 0.001), especially in males [20]. In additional analyses, fibre intake reduced the allergic rhinitis symptoms, including watery rhinorrhoea and dog allergen sensitisation, only among males. However, in a cross-sectional study of 1002 Japanese pregnant women from the Osaka Maternal and Child Health Study, no association between fibre intake and allergic rhinitis was reported [21].

A French cross-sectional study of 26,640 women and 8740 men reported inverse associations between the highest quintile of total dietary fibre compared with the lowest quintile and the asthma symptom scores both among women (OR 0.73; 95% CI 0.67–0.79, *p*-trend < 0.001) and men (OR 0.63; 95% CI 0.55–0.73, *p*-trend < 0.001) [22]. With regards to specific sources of fibre, the intake of fibre from cereals, fruit and seeds was most consistently associated with less asthma symptoms. Additionally, among participants with asthma, inverse associations were reported between the fibre intake and uncontrolled asthma. In an Australian case–control study of 137 participants with asthma and 65 healthy controls, participants with severe persistent asthma (*n* = 64) consumed, on average, 5 g/day less fibre as compared to healthy controls (OR 0.94; 95% CI 0.90–0.99) [23].

##### COPD and Related Symptoms

Eight studies reporting the fibre intake in adulthood in relation to COPD and COPD symptoms were identified, with consistent results of a protective association. A cross-sectional study of 11,897 participants of the Atherosclerosis Risk in Communities (ARIC) study in the US indicated a reduced prevalence of COPD with a higher total fibre intake (Q5 vs. Q1, OR 0.85; 95% CI 0.68–1.05, *p*-trend = 0.044). Inverse associations were also observed with cereal or fruit fibre but not with vegetable fibre [24]. No interaction with smoking status was observed, although associations were limited to current or ex-smokers. In a Japanese case–control study, high levels of total and insoluble dietary fibre were associated with a reduced risk of COPD (Q4 vs. Q1, OR 0.49; 95% CI 0.26–0.95, *p*-trend 0.160 and OR 0.50; 95% CI 0.26–0.94, *p*-trend 0.174, respectively) [25].

A study of 49,140 cohort members from the Singapore Chinese Health Study examining the association between dietary fibre and new onset of cough with phlegm reported inverse associations with non-starch polysaccharides (Q4 vs. Q1, OR 0.61; 95% CI 0.47–0.78, *p*-trend < 0.001), fruits (OR 0.67; 95% CI 0.52–0.87, *p*-trend 0.006) and soy isoflavones (OR 0.67; 95% CI 0.53–0.86, *p*-trend 0.001) [26]. Moreover, a large cohort study of 111,580 participants from the US Nurses’ Health Study and Health Professionals Follow-up Study with long follow-up periods (16 and 12 years, respectively) reported inverse associations between the total dietary fibre intake and newly diagnosed COPD (Q5 vs. Q1, RR 0.67; 95% CI 0.50–0.90, *p*-trend 0.03) [27]. Inverse associations were also observed with cereal fibre but not with fruit or vegetable fibre. In stratified analyses by the smoking status, associations were stronger among current smokers than among ex-smokers. Two cohort studies on dietary fibre intake from Sweden, which used registry data to identify incident COPD cases, confirmed the aforementioned results. The first study included 45,058 men from the Cohort of Swedish Men and reported strong inverse associations with the total fibre intake (Q5 vs. Q1, HR 0.62; 95% CI 0.53–0.71, *p*-trend < 0.001), mainly in current smokers or ex-smokers but not in never smokers [28]. The second study included 35,339 women from the Swedish Mammography Cohort and evaluated the association between the baseline and long-term total fibre intake and COPD risk; in this study, a high long-term dietary fibre intake was associated with a reduced risk of COPD (Q5 vs. Q1, HR 0.70; 95% CI 0.59–0.83, *p*-trend < 0.001), mainly in current or ex-smokers. For specific fibre sources, cereal and fruit fibre, but not vegetable fibre, were associated with a lower COPD risk [29].

A cross-sectional study of 702 adults with COPD from the KNHANES evaluated the association between disease severity and dietary nutrient intake; in this study, fibre intake was associated with a decreased severity of airway impairment in elderly men (≥60 years old) with COPD but not in women [30]. Additionally, a cohort study of 1439 participants from Korea studied the relationship between new airflow limitation development, defined as FEV1/FVC < 0.70, and changes to the dietary pattern after a 5-year period; in this study, a 10% decreased intake of dietary fibre was associated with a newly developed airflow limitation (OR 2.71; 95% CI 1.54–4.81) [31].

##### Lung Function

Six studies reporting fibre intake in adulthood in relation to lung function were identified, with generally consistent findings. The already mentioned study of 11,897 participants from the ARIC study in the US found positive cross-sectional associations between the total fibre and lung function (Q5 vs. Q1, forced expiratory volume in one sec (FEV1) 60.2 mL; 95% CI 27.7–92.7, *p*-trend < 0.001, forced vital capacity (FVC) 55.2 mL; 95% CI 18.2–92.3, *p*-trend 0.001 and FEV1/FVC 0.4; 95% CI −0.1–0.9, *p*-trend 0.040). Similar patterns were seen for the fibre intake from cereal and fruit sources, while no association was observed for vegetable fibre [24]. Additionally, a recent cross-sectional study of 1921 participants from the US NHANES examined the association between fibre intake and measures of lung function. According to this study, a low fibre intake was associated with reduced measures of lung function (Q4 vs. Q1, FEV1 82 mL (*p* = 0.05), FVC 129 mL (*p* = 0.01), % predicted FEV1 2.4% (*p* = 0.07) and % predicted FVC 2.8% (*p* = 0.02)) [32].

Another prospective study of 12,532 adults from the ARIC study reported an increased fibre intake associated with improved lung function when followed up three years after the baseline; the coefficients per increase in one quintile of fibre intake were %FEV1 0.201, *p*-trend ≤ 0.05 and FEV1/FVC 0.129, *p*-trend ≤ 0.01, but there was no association with FEV1 or FVC [33]. A prospective study including 5880 participants from the Korean Ansan-Ansung cohort followed for four years indicated a positive association between the fibre intake and lung function among men but not among women [34]. A study among smokers in the US Lovelace Smokers cohort (LSC), with replication in the Veteran Smokers cohort (VSC), identified, among other nutrients, the fibre intake to be significantly associated with a better average FEV1 (LSC 80.9 mL; SE 20.3, *p* = 0.0032 and VSC 97.8 mL; SE 41.8, *p* = 0.045) [35]. Finally, in cross-sectional analyses in the aforementioned Australian case–control study of 137 participants with asthma and 65 healthy controls, the fibre intake was positively associated with FEV1, FVC and FEV1/FVC (coefficient per unit increase in fibre intake 0.02 L (*p* = 0.001), 0.02 L (*p* = 0.002) and 0.2% (*p* = 0.035), respectively) and negatively associated with airway eosinophilia (−0.36% (*p* = 0.005)) among participants with asthma [23].

### 3.4. Quality of Studies

#### 3.4.1. Asthma and Related Symptoms

Out of seven studies, five were cross-sectional [17,18,19,20,22], one was a case–control [23] and one a cohort study [16]. With regards to fibre intake assessment, four studies used food frequency questionnaires (FFQ) [16,17,20,23], and three studies used 24-h dietary recalls [18,19,22], including repeated assessments in the last two studies. Regarding the outcome assessment, asthma was self-reported in all studies, while five studies also included a clinical examination with spirometry, skin prick tests and/or blood sampling [16,17,19,20,23]. All studies adjusted for age and sex, while most studies adjusted for body mass index (BMI) or energy intake, socioeconomic factors and smoking. Overall, the articles were assigned a moderate-to-serious risk of bias (Figure 2a).

#### 3.4.2. Rhinitis and Related Symptoms

Both studies had a cross-sectional study design [20,21]. To assess the fibre intake, both studies used a FFQ [20,21]. Allergic rhinitis was self-reported in both studies and assessed based on the symptoms and, additionally, nasal endoscopy and serum IgE levels in one study [20], while the assessment was based on drug treatment in the previous 12 months in the other study [21]. Both studies adjusted for major potential confounders, including age, BMI, socioeconomic factors and smoking. Overall, the articles were assigned a moderate-to-serious risk of bias (Figure 2b).

#### 3.4.3. COPD and Related Symptoms

Out of eight studies, five were cohorts [26,27,28,29,31], two were cross-sectional [24,30] and one a case–control study [25]. Fibre intake was assessed using FFQs in all but one study [24,25,26,27,28,29,31], with repeated assessments in three studies [27,29,31] and one study using a 24-h dietary recall [30]. COPD and related symptoms were assessed using self-reported questionnaires in three studies [24,26,27], spirometry in four studies [24,25,30,31] and registries in two studies [28,29] (one study used both self-reported and spirometry diagnosed definitions). All the studies adjusted for major potential confounders, including age, sex and smoking, and most adjusted for BMI or energy intake and socioeconomic factors, while some additionally adjusted for lifestyle (physical activity and alcohol intake) and other dietary factors. Overall, the articles were assigned a moderate risk of bias (Figure 2c).

#### 3.4.4. Lung Function

Out of six studies, there were three cohort [33,34,35] and three cross-sectional studies [23,24,32]. Fibre intake was assessed using FFQs in all studies, apart from one study that used repeated 24-h dietary recalls [32]. Lung function was measured by spirometry in all the studies and additionally using eNO and combined bronchial provocation and sputum induction in one study [23]. Most of the studies adjusted for major potential confounders, including age, sex, BMI, total energy intake, smoking and socioeconomic factors. Overall, the articles were assigned a moderate risk of bias (Figure 2d).

### 3.5. Strength of the Evidence

In the present study, we did not include intervention studies in the eligibility criteria. According to the WCRF criteria, and based on the available evidence from observational studies, the overall strength of the evidence was graded as limited/no conclusion (insufficient) with regards to asthma and rhinitis, probable (moderate) for COPD and limited/suggestive (low) for lung function.

## 4. Discussion

This review sought to explore if there is a protective association between dietary fibre intake and asthma, rhinitis, COPD and lung function and, if so, which sources of fibre are the most beneficial. The findings show that the current evidence from observational studies is limited and inconclusive with regards to asthma and rhinitis. There is suggestive evidence that dietary fibres may be associated with improved lung function in the general adult population, with very few studies reporting fibre intake in high-risk populations. Moreover, there is probable evidence for a beneficial role of fibres in the risk of COPD, which is considered as strong evidence according to the WCRF criteria.

Based on the intake level observed to protect against coronary heart disease, an adequate intake of total fibre has been set to 30–35 g/day and 25–32 g/day for adult men and women, respectively [36], and 10–40 g for children and adolescents, depending on age, gender and energy intake [37]. The mean fibre intake was reported to be below the recommended levels in all the included studies, and only subjects in the highest quartile/quintile of fibre intake met the recommendations. Additionally, geographical differences in the amount of total dietary fibre intake were observed, with lower intakes reported in studies from countries in Asia, followed by the US, and higher intakes in studies from countries in Australia and Europe. With regards to different types of diets, while the fibre content of animal products is scarce, plant-based diets include fibre-rich foods, such as cereals, fruits, vegetables and nuts, in abundance. We observed a difference in the sources of dietary fibres in the studied populations as well, reflecting different dietary patterns; however, this was not consistently reported in all the included studies. The current evidence is not conclusive about which fibre-rich foods are most beneficial for respiratory health; the grain sources of dietary fibre have been shown to be more beneficial compared to fruits and vegetables, but it is unclear if this is due to their higher fibre content, greater amounts consumed, less probability of measurement error, displacement of high-energy foods and overall diet quality or associated lifestyle factors, such as greater levels of physical activity [36].

We were able to identify only one study on maternal fibre intake during pregnancy in relation to allergic disease in the offspring. Although these results do not support an association between prenatal exposure to dietary fibre and allergic disease, the association is biologically plausible. In two recent birth cohort studies, the faecal concentration of SCFAs during pregnancy was inversely associated with asthma and allergic rhinitis in the offspring up to 6 years [38,39]. In one of the studies, which was part of a randomised controlled trial and, therefore, not included in the selection criteria of our review, the fibre intake in pregnancy was positively associated with the total SCFAs but not with any of the atopic outcomes in the offspring [38]. It was therefore hypothesised that dietary fibres contribute to offspring disease risk only in combination with the relevant intestinal microbes. These findings, supported by studies in animal models [39,40], require further replication in observational studies with a larger sample size and can potentially pave the way to microbiome-targeted interventions to prevent asthma and atopy in the offspring [9,41,42].

Additionally, the two selected studies on dietary fibre intake during childhood showed inconsistent results. However, previous reviews have reported protective associations between fruit and vegetable consumption, or dietary patterns rich in fruits, vegetables, legumes and cereals (such as the Mediterranean diet), and asthma or wheeze among children [43,44,45]. Moreover, a protective association between whole grains and asthma among children has been reported [46]. These associations may be partly explained by the concomitant intake of dietary fibres. Dietary fibres can potentially improve airway inflammation by promoting anti-inflammatory cytokines, improving glucose control, and modulating the gut immunologic response [10]. On the other hand, asthma is a heterogeneous disease, and asthma development is a dynamic process, characterised by remission, relapse and a new onset of symptoms from childhood up to adulthood [47]. A reduced fibre intake has been observed among adults with severe asthma and has been associated with increased eosinophilic airway inflammation [23]. Lung function growth may not only be impaired during early childhood but also continues throughout adolescence and early adulthood [48]. Thus, a critical period of development is missed by the current body of evidence, addressing dietary fibre intake and asthma and lung function impairment in adult populations. A paucity of studies addressing asthma severity, different asthma phenotypes and lung function among participants with asthma has also been identified.

We were able to identify only two studies on fibre intake in relation to allergic rhinitis, with inconsistent results, and no study on nonallergic rhinitis. Allergic rhinitis is associated with sensitisation to inhalant allergens, whereas nonallergic rhinitis is a nasal mucosal inflammation without systemic signs of allergic inflammation, associated with exposure to irritants, hormonal dysfunction and specific medications [49,50]. Regarding allergic rhinitis, a high fibre intake in a murine model showed less eosinophil infiltration, less goblet cell metaplasia in the nasal mucosa and decreased Th2 cytokines compared to a low intake [51]. In a study among children, adherence to the Mediterranean diet has been inversely associated with allergic rhinitis [52]. Further research is needed on dietary fibre intake and rhinitis outcomes, both among children and adults.

In our review, we identified eight studies on fibre intake in relation to COPD and related symptoms reporting consistent results of a protective association, which has also been suggested by two previous systematic reviews of studies on fibre intake in relation to COPD, partly based on the same studies [53,54]. The association with COPD may be explained by the antioxidant and anti-inflammatory properties of dietary fibres, including lower levels of C-reactive protein and proinflammatory cytokines and higher levels of some anti-inflammatory cytokines, such as adiponectin [54]. In addition, high dietary fibre has been suggested to attenuate innate immune-mediated systemic and pulmonary inflammation through the presence of a gut–liver–lung axis [55]. The stronger inverse association with COPD among current or ex-smokers may be explained by the higher oxidative stress in these groups, as well as the continued endogenous production of reactive oxygen species even after smoking cessation. Among non-smokers, the mechanisms related to COPD development may differ from those in current or ex-smokers and relate more to genetic predisposition and environmental exposures [28]. Among smokers, lung function was improved via the increased intake of dietary fibre, further supporting the importance of the gut–liver–lung axis in COPD [41]. On the other hand, a protective effect of fibre intake on lung function in both smokers and non-smokers has also been observed [24]. In non-smokers, fibre intake may protect against the deleterious effects of indoor and ambient air pollutants. Considering the sources of dietary fibres, the results from the included studies suggest that fibres from cereals and fruits, but not vegetables, are inversely associated with the risk of COPD. It has been suggested that similar protective associations of a higher intake of cereal fibre and of the total dietary fibre may be because of the high dietary fibre content of cereals [56]. Nevertheless, the lack of an inverse association with vegetables has been suggested to be related to the higher uptake of heavy metals, especially cadmium and lead, from vegetables compared to fruits [29].

The strength of evidence is primarily related to the methodological quality of the included studies. Most of the studies used questionnaires to assess dietary fibre intake, which might have led to some misclassification of the exposure, however nondifferential with regards to the outcomes of interest. Although the absolute fibre intake may be difficult to be estimated by FFQs, the ranking of participant intakes is possible and sufficient in this type of analytic epidemiologic studies [57]. In studies where multiple exposures were studied or fibre was part of an overall dietary pattern, the methods used to assess the exposure were less well-described, which hampers systematic reviews and meta-analyses [58]. In line with this, an increased risk of selective reporting could be inferred from studies reporting associations with multiple outcomes and analyses among different subgroups [13]. The risk of publication bias for studies finding no associations between fibre intake and respiratory and atopic outcomes should also be considered. Although most of the studies extensively adjusted their analyses for major potential confounders and some additionally included dietary and lifestyle factors, such as smoking, physical activity and alcohol consumption, residual confounding cannot be completely ruled out. In this systematic review, we were able to assess the strength of the current evidence based on observational studies and highlight specific areas where further research is needed.

## 5. Conclusions

In conclusion, the current evidence from observational studies on dietary fibre intake is probable (moderate) for an inverse association with COPD and limited/suggestive (low) for an association with lung function in the general adult population. In contrast, there is insufficient evidence for an association with asthma or rhinitis in adults. Thus, further research is needed with regards to asthma, rhinitis and lung function in adults, as well as among children.

## Figures and Tables

**Figure 1 nutrients-13-03594-f001:**
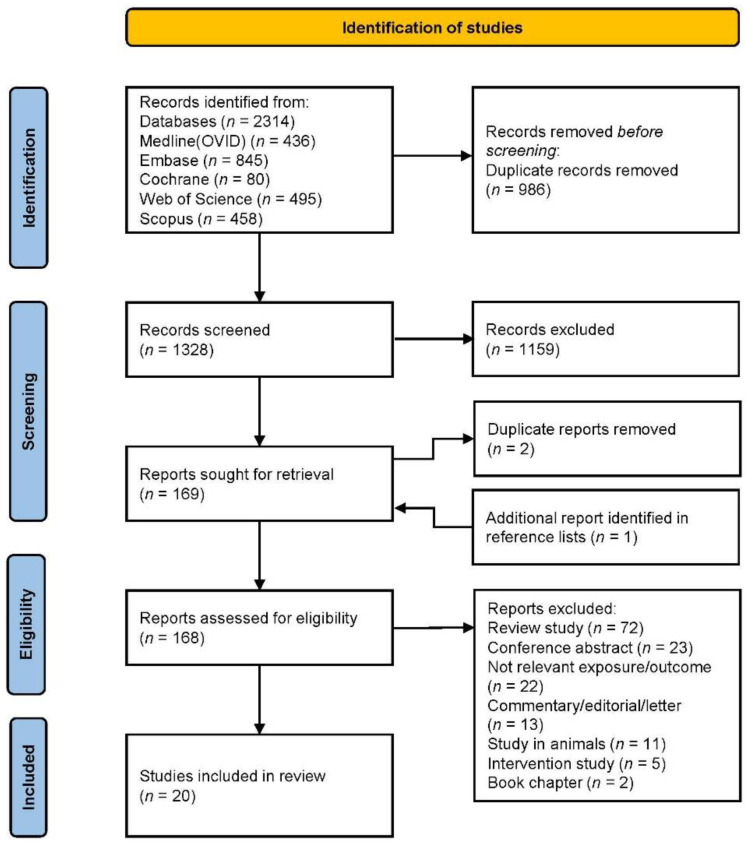
PRISMA flowchart of the study selection.

**Figure 2 nutrients-13-03594-f002:**
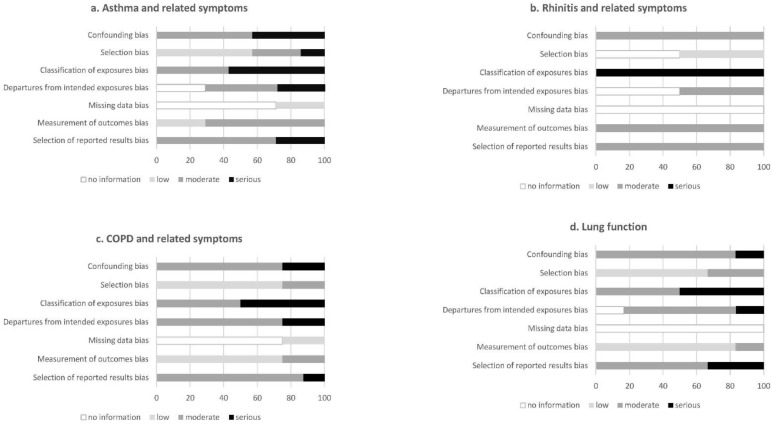
Risk of bias in individual studies, showing the proportion of studies with no information and low, moderate and serious risks of bias in each domain of the RoB-NObs tool.

**Table 1 nutrients-13-03594-t001:** Characteristics of the studies examining the association between dietary fibre intake and asthma, rhinitis, COPD, and lung function.

Author, Year, Country, Cohort	Study Design	Sample Size	Population	Exposure Assessment	Outcome Assessment	Follow-Up	Statistical Methods	Effect Measures	Covariates
**Maternal fibre intake during pregnancy**									
Pretorius et al., 2019, Australia [16]	C	639 mother–infant pairs	mothers aged ≥ 18 years, non-smokers, infants with family history of allergic disease	semi-quantitative 101-item FFQ at 36–40 weeks’ gestation assessing intake over one month	parent reported and doctor diagnosed wheeze	12 months	logistic regression, multinomial logistic regression	total fibre (g/day), median 23.8 OR (95% CI) 0.99 (0.99–1.00) parent reported wheeze OR (95% CI) 0.98 (0.94–1.01) doctor diagnosed wheeze	maternal age, education, ethnicity, child’s gender, birth weight, gestational age at birth, pet ownership, maternal parity, delivery mode
**Fibre intake during childhood**									
Wood et al., 2015, Australia [17]	CS	144	adolescents aged 12–18 years	interviewer administered 107-item FFQ	ISAAC video questionnaire, spirometry	-	logistic regression	total fibre (g/day) OR (95% CI) 1.0 (1.0–1.0) self-reported wheeze	age, sex, length of time in Australia
Vaccaro et al., 2016, US, NHANES [18]	CS	4133	children aged 2–11 years	24 h dietary recall	self-reported asthma	-	logistic regression	energy adjusted fibre, median 6.7 g/1000 kcal Q1 vs. Q4 OR (95% CI) 1.31 (0.88–1.96), *p*-trend 0.034 ever asthma, Q1 vs. Q4 OR (95% CI) 1.38 (0.87–2.20), *p*-trend 0.027 current asthma	age, gender, second-hand smoke exposure, income
**Fibre intake during adulthood**									
* **Asthma, rhinitis and related symptoms** *									
Saeed et al., 2020, US, NHANES [19]	CS	13,147	adults aged 20–79 years (mean age 46 years)	two-interviewer-administered 24 h recalls	self-reported asthma, wheeze, cough, phlegm production, blood CRP	-	logistic regression, multinomial logistic regression	total fibre Q1 vs. Q4 (<10.5 vs. >21.2 g/day) OR (95% CI) 1.4 (1.0–1.8), *p*-trend 0.092 asthma, OR (95% CI) 1.3 (1.0–1.6), *p*-trend 0.017 wheeze, OR (95% CI) 1.7 (1.2–2.3), *p*-trend 0.0003 cough, OR (95% CI) 1.4 (1.1–2.0), *p*-trend 0.011 phlegm	age, race/ethnicity, sex, smoking status, BMI, poverty index ratio, total energy intake
Lee et al., 2021, South Korea, Korean NHANES [20]	CS	10,479	adults aged 19 years and older (mean age 51 years)	63-item FFQ	self-reported asthma, self-reported rhinitis plus nasal endoscopy, serum IgE and specific IgE levels	-	logistic regression	total fibre Q4 vs. Q1 OR (95% CI) 0.66 (0.48–0.91), *p*-trend < 0.001 asthma, OR (95% CI) 0.95 (0.77–1.17), *p*-trend < 0.001 allergic rhinitis	age, sex, household income, residency, alcohol consumption, smoking status, BMI, physical activity, other nutrients
Miyake et al., 2006, Japan, Osaka Maternal and Child Health Study [21]	CS	1002	pregnant women	147-item questionnaire assessing intake over one month	allergic rhinitis drug treatment during the previous 12 months	-	logistic regression	energy adjusted fibre (g/day) Q4 (14.7) vs. Q1 (8.2) OR (95% CI) 1.14 (0.66–2.00), *p*-trend 0.80	age, gestation, parity, cigarette smoking, passive smoking, indoor domestic pets, family history of asthma, eczema, rhinitis, family income, education, mite allergen level in house dust, changes in diet in the previous month, season of data collection, BMI
Andrianasolo et al., 2019, France, NutriNet-Sante Study [22]	CS	26,640 women and 8740 men	adults aged 18 years and older (mean age 53 years in women, 59 years in men)	three self-administered web-based 24 h dietary records	self-reported asthma symptom score, asthma control test	-	binomial negative regression, logistic regression	total fibre Q5 vs. Q1 (28.6 vs. 13.8 g/day in women, 30.5 vs. 12.7 g/day in men) asthma symptom score OR (95% CI) 0.73 (0.67–0.79), *p*-trend < 0.001 in women and OR (95% CI) 0.63 (0.55–0.73), *p*-trend < 0.001 in men	age, educational level, smoking status, pack-years of smoking, BMI, physical activity, total energy intake, allergic rhinitis, family history of asthma
Berthon et al., 2013, Australia [23]	CC	137 cases with asthma (of which 64 with severe persistent asthma), 65 controls	adults aged 18 years and older (mean age 53 years)	186-item semi-quantitative FFQ	asthma severity, lung function (eNO, spirometry, sputum cells)	-	logistic and linear regression	energy adjusted fibre (mean cases 32 g/day, controls 37 g/day) severe persistent asthma OR (95% CI) 0.94 (0.90–0.99) Coefficients for dietary fibre intake in asthmatics FEV1 0.02 L, FVC 0.02 L, FEV1/FVC 0.002, airway % eosinophils −0.36, % neutrophils 0.26	age, gender, BMI, total energy
* **COPD and related symptoms** *									
Kan et al., 2008, US, ARIC study [24]	CS	11,897	adults aged 44–66 years	interviewer-administered 66-item semi-quantitative FFQ	spirometry, COPD based on self-reported symptoms or spirometry	3 years	linear regression, logistic regression	total fibre (g/day) Q5 (26.7) vs. Q1 (9.5) OR (95% CI) COPD prevalence 0.85 (0.68–1.05), *p*-trend 0.044 Coefficients (95% CI) FEV1 60.2 mL (27.7–92.7), *p*-trend < 0.001, FVC 55.2 mL (18.2–92.3), *p*-trend 0.001, FEV1/FVC 0.4 (−0.1–0.9), *p*-trend 0.040, % predicted FEV1 1.8 (0.8–2.9), *p*-trend < 0.001, % predicted FVC 1.4 (0.4–2.4), *p*-trend 0.001	BMI, age, ethnicity, gender, study centre, smoking status, pack-years, occupation, education, diabetes status, residence-based traffic density, total energy intake, glycaemic index, micronutrients from both food and supplements, and cured meat
Hirayama et al., 2009, Japan [25]	CC	278 cases with COPD, 340 controls	adults aged 50–75 years (mean age 66 years)	138-item FFQ assessing intake over the previous five years	spirometry diagnosed COPD within the past four years	-	unconditional logistic regression	total fibre (g/day) Q4 (≥16.08) vs. Q1 (≤8.84) OR (95% CI) 0.49 (0.26–0.95), *p*-trend 0.160	age, gender, BMI, education level, life-long physical activity involvement, smoking status, smoking pack-years, alcohol drinking, intake of fish, red meat and chicken, total energy intake
Butler et al., 2004, Singapore, Singapore Chinese Health Study [26]	C	49,140	adults aged 45–74 years of Chinese origin	165-item quantitative FFQ	self-reported incident cough with phlegm	5.3 years	unconditional logistic regression	Q4 (11.6) vs. Q1 (4.7) OR (95% CI) non-starch polysaccharides (g) 0.61 (0.47–0.78), *p*-trend 0.001, fruit 0.67 (0.52–0.87), *p*-trend 0.006, grain 1.12 (0.80–1.56), *p*-trend 0.301, vegetable 0.92 (0.70–1.21), *p*-trend 0.504, soy isoflavones 0.67 (0.53–0.86), *p*-trend 0.001	age, sex, dialect group, total energy intake, smoking status, age of smoking initiation, amount smoked
Varraso et al., 2010, US, Nurses’ Health Study, Health Professionals Follow-up Study [27]	C	111,580	female nurses aged 30–55 years, men health professionals 40–75 years, no history of asthma or COPD	FFQs administered in 1984, 1986, 1990, 1994 and 1998 in NHS and in 1986, 1990 and 1994 in HPFS	self-reported COPD defined by doctor diagnosis of chronic bronchitis or emphysema and diagnostic test	16 and 12 years	Cox proportional hazard regression models	total fibre Q5 (28.4) vs. Q1 (11.2) RR (95% CI) 0.67 (0.50–0.90), *p*-trend 0.03	age, sex, smoking status, pack-years, pack-years2, energy intake, BMI, US region, physician visits, physical activity, diabetes, intakes of omega-3 (foods and supplements), cured meat, (glycaemic index, carotenoids, vitamins C, D, E)
Kaluza et al., 2018, Sweden, Cohort of Swedish Men [28]	C	45,058	men aged 45–79 years, no history of COPD	96-item FFQ	incident COPD cases through linkage with registry data	13.1 years	Cox proportional hazard regression models	total fibre (g/day) Q5 (≥36.8) vs. Q1 (<23.7) HR (95% CI) 0.62 (0.53–0.71), *p*-trend < 0.001	age, education, BMI, total physical activity, smoking status, pack-years of smoking, alcohol intake, energy intake
Szmidt et al., 2020, Sweden, Swedish Mammography Cohort [29]	C	35,339	women aged on average 62 years, no history of COPD	67-tem FFQ in 1987, 96-item FFQ in 1997 (baseline)	incident COPD cases through linkage with registry data	11.5 years	Cox proportional hazard regression models	long-term total fibre (g/day) Q5 (≥26.5) vs. Q1 (<17.6) HR (95% CI) 0.70 (0.59–0.83), *p*-trend < 0.001	age, education, BMI, walking/cycling, smoking status, pack-years of smoking, alcohol, energy intake
Kim et al., 2019, South Korea, Korean NHANES [30]	CS	702	COPD adults aged ≥ 40 years	24 h dietary recall	COPD severity defined by spirometry	-	linear regression	mean (SE) total fibre (g/day) severity men 20.9 (1.7), women 18.3 (1.8)	sex, age, residential area, educational level, household income, smoking status, height
Jung et al., 2021, Korea [31]	C	1439	adults aged on average 53 years	117-item FFQ assessing intake over the previous three months, in 2012 and 2017	incident COPD cases defined by spirometry	5 years	logistic regression	decrease in total fibre (g/day) Q4 vs. Q1 decrease in total fibre, proportion of new airflow limitation cases 5.85% vs. 1.39%	age, sex, smoking history, baseline FEV1/FVC
* **Lung function** *									
Hanson et al., 2016, US, NHANES [32]	CS	1921	adults aged 40–79 years (mean age 53 years)	two-interviewer administered 24 h recalls	spirometry	-	regression analyses	total fibre (g/day) Q4 (<10.75) vs. Q1 (>17.5) Coefficients FEV1 82 mL, FVC 129 mL, % predicted FEV1 2.4, % predicted FVC 2.8	age, sex, smoking status, height, BMI, socioeconomic status, total energy intake, CRP, vitamin E, a-carotene, b-carotene, b-cryptoxanthin, lycopene, lutein plus zeaxanthin, vitamin C and cured meat
Root et al., 2014, US, ARIC study [33]	C	12,532	adults aged on average 54 years	interviewer-administered semi-quantitative FFQ	spirometry	3 years	linear regression	Coefficients per increase in one quintile of total fibre FEV1 NS, FVC NS, %FEV1 0.201, *p*-trend ≤ 0.05, FEV1/FVC 0.129, *p*-trend ≤ 0.01	age, sex, ethnicity, education, total caloric intake, physical activity, current smoking, cigarette years, height, BMI, and interaction term black ethnicity × BMI
Lee et al., 2020, Korea, Ansan-Ansung cohort [34]	C	5880	non-COPD adults median age 50 years	103-item FFQ	spirometry (% difference of predicted FEV1 between baseline and follow-up)	4 years	logistic regression	Q5 (≥8.9) vs. Q1 (≤4.4 g/day) decreased vs. unchanged/improved OR (95% CI) 0.83 (0.61–1.12), *p*-trend 0.080 in men, 1.14 (0.86–1.51), *p*-trend 0.345 in women	age, education, household income, job, BMI, waist circumference, waist-to-hip ratio, smoking, alcohol, exercise, marriage status, history of asthma and tuberculosis, energy intake
Leng et al., 2017, US, Lovelace Smokers cohort (LSC), Veteran Smokers cohort (VSC) [35]	C	1829 in LSC, 508 in VSC	adult smokers aged 40–74 years (mean age 57 in LSC, 62 in VSC)	semi-quantitative 150-item FFQ	spirometry	5.3 years	linear mixed effects model with a subject-specific random intercept, linear regression	total fibre (g/day), mean 10.5 Coefficients (SE) FEV1 LSC 80.9 mL (20.3), VSC 97.8 mL (41.8), FEV1/FVC% LSC 1.075 (0.403), VSC 2.018 (0.761)	age, sex, ethnicity, smoking history, BMI, educational level, height, total caloric intake, time since enrolment, baseline FEV1 in decline analysis

Abbreviations: C: cohort, CC: case–control, CS: cross-sectional, OR: odds ratio, RR: risk ratio, HR: hazard ratio, CI: confidence interval, SE: standard error, IQR: interquartile range, FFQ: food frequency questionnaire, BMI: body mass index, COPD: chronic obstructive pulmonary disease, FEV1: forced expiratory volume in one second, FVC: forced vital capacity, NS: nonsignificant, NHANES: National Health and Nutrition Examination Survey, ARIC: Atherosclerosis Risk in Communities, LSC: Lovelace Smokers Cohort, and VSC: Veteran Smokers Cohort.

## Data Availability

Data is contained within the article or Appendix A.

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
