# Peer review of "Dietary Fibre Intake in Relation to Asthma, Rhinitis and Lung Function Impairment—A Systematic Review of Observational Studies"

_nutrients, 2021, doi:10.3390/nu13103594_

Round 1
Reviewer 1 Report
In this systematic review, Sdona and collaborators abroad a review of existing literature about dietary fiber intake and how it affects to development of asthma, rhinitis and other lung diseases. There is not a lot of studies about this topic and finally do not exist a sufficient evidence. Nevertheless, this study is well conducted and more studies in this field should be performed. This systematic review is interesting and relevant in the field. I only have few comments about it:
- Authors have proved evidences that fiber intake can improve lung function in patients with chronic respiratory diseases, based in previous articles. However, readers could found very interesting (if it is possible) a section where different food (fruits, vegetables, cereals, etc.) classified by its quantity of fiber would be able to related with effects in respiratory diseases. It is possible that food with more percentage of fiber (or kind of fiber) could improve in a different manner the resolution or improvement of respiratory diseases.
Author Response
We thank the Reviewer for reviewing our manuscript and for the comments. Following the Reviewer’s suggestion, we added the following in the Discussion (lines 357-363): “Current evidence is not conclusive about which fibre-rich foods are most beneficial for respiratory health; grain sources of dietary fibre have been shown to be more beneficial compared to fruits and vegetables, but it is unclear if this is due to their higher fibre content, greater amounts consumed, less probability of measurement error, displacement of high-energy foods and overall diet quality, or associated lifestyle factors, such as greater levels of physical activity (Stephen et al. 2017).” Additionally, regarding associations with COPD, which are more consistent, we report the following in the Discussion (lines 423-430): “Considering sources of dietary fibres, results from included studies suggest that fibres mainly from cereals, followed by fruits, but not vegetables, are inversely associated with risk of COPD. It has been suggested that the similar protective associations of higher intake of cereal fibre and of total dietary fibre may be because of the high dietary fibre content of cereals (Reynolds et al. 2019). Nevertheless, the lack of an inverse association with vegetables has been suggested to be related to the higher uptake of heavy metals, especially cadmium and lead, from vegetables compared to fruits (Szmidt et al. 2020).”
Reviewer 2 Report
Authors wished to review the current evidence on the association between dietary fibre and asthma, rhinitis, chronic obstructive pulmonary disease (COPD) and lung function through a systematic review of observational studies with cross-sectional (8 studies), case-control (2 studies) or prospective (10 cohort studies) design.
What they found is a probable evidence for an inverse association between dietary fibre and COPD, and suggestive evidence for a positive association with lung function, with no evidence of association between fiber intake and asthma and rhinitis .
Critical points
- The major risk factor for COPD is smoking history and pollutants exposure. How fiber dietary intake may blunt the effects of smoking should be discussed
- Asthma is a heterogenous disease. It would be interesting to explore the different effects that dietary fiber intake may have on eosinophilic, allergic and non eosinophilic asthma phenotypes.
-Rhinitis too is a heterogenous disease and Authors should consider separately allergic and non allergic rhinitis.
Author Response
Comment 1: The major risk factor for COPD is smoking history and pollutants exposure. How fibre dietary intake may blunt the effects of smoking should be discussed.
Answer: We thank the Reviewer for reviewing our manuscript and for this comment. Studies included in the systematic review on fibre intake in relation to COPD that examined effect modification by smoking status found somewhat consistent results of a stronger inverse association among current or ex-smokers. Following the Reviewer’s comment, we added in the Results: (lines 215-216) “No interaction with smoking status was observed, although associations were limited to current or ex-smokers (Kan et. Al 2008)”, (lines 230-231) “In stratified analyses by smoking status, associations were stronger among current smokers than among ex-smokers (Varraso et al. 2010)”, and in the Discussion: (lines 413-423) “The stronger inverse association with COPD among current or ex-smokers may be explained by the higher oxidative stress in these groups, as well as continued endogenous production of reactive oxygen species even after smoking cessation. Among non-smokers, the mechanisms related to COPD development may differ from those in current or ex-smokers and relate more to genetic predisposition and environmental exposures (Kaluza et al. 2018). Among smokers, lung function was improved via increased intake of dietary fibre, further supporting the importance of gut microbiota and gut-liver-lung axis in COPD (Zhang et al. 2020). On the other hand, a protective effect of fibre intake on lung function in both smokers and non-smokers has also been observed (Kan et al. 2008). In non-smokers, fibre intake may protect against the deleterious effects of indoor and ambient air pollutants.”
Comment 2: Asthma is a heterogenous disease. It would be interesting to explore the different effects that dietary fiber intake may have on eosinophilic, allergic and non-eosinophilic asthma phenotypes.
Answer: We agree with the Reviewer that it would be interesting to explore dietary fibre intake in relation to the different asthma phenotypes. However, relevant evidence from observational studies is currently limited. Following the Reviewer’s comment, we modified the paragraph on asthma in the Discussion (lines 383-394), as follows: “Dietary fibres can potentially improve airway inflammation by promoting anti-inflammatory cytokines, improving glucose control, and modulating gut immunologic response (Alwarith et al. 2020). On the other hand, asthma is a heterogeneous disease and asthma development is a dynamic process throughout childhood, characterized by remission, relapse and new onset of symptoms up to adulthood (Fuchs et al. 2017). Reduced fibre intake has been observed among adults with severe asthma and has been associated with increased eosinophilic airway inflammation (Berthon et al. 2013)… A paucity of studies addressing asthma severity, different asthma phenotypes, and lung function among participants with asthma has also been identified.”
Comment 3: Rhinitis too is a heterogenous disease and Authors should consider separately allergic and non-allergic rhinitis.
Answer: Following the Reviewer’s comment, we added an extra paragraph in the Discussion on rhinitis (lines 395-404), as follows: ”We were able to identify only two studies on fibre intake in relation to allergic rhinitis, with inconsistent results, and no study on non-allergic rhinitis. Allergic rhinitis is associated with sensitization to inhalant allergens, whereas non-allergic rhinitis is a nasal mucosal inflammation without systemic signs of allergic inflammation, associated with exposure to irritants, hormonal dysfunction, and specific medications (Roberts et al. 2013, Hellings et al. 2017). Regarding allergic rhinitis, high fibre intake in a murine model showed less eosinophil infiltration, less goblet cell metaplasia in the nasal mucosa, and decreased Th2 cytokines compared to low intake (Zhang et al. 2016). In a study among children, adherence to the Mediterranean diet has been inversely associated with allergic rhinitis (Chatzi et al. 2007). Further research is needed on dietary fibre intake and rhinitis outcomes, both among children and adults.”